# Assessing Patterns of Anti-Social and Risky Behaviour in the Millennium Cohort Study—What Are the Roles of SES (Socio-Economic Status), Cognitive Ability and Personality?

**DOI:** 10.3390/bs13010046

**Published:** 2023-01-04

**Authors:** Michael O’Connell

**Affiliations:** School of Psychology, University College Dublin (UCD), Belfield, D04 V1W8 Dublin, Ireland; michael.f.oconnell@ucd.ie

**Keywords:** adolescents, anti-social behaviour, SES, cognitive ability, personality measures, hyperactivity

## Abstract

Data from the Millennium Cohort Study (UK) were examined to assess the correlates of anti-social and risky behaviour among adolescents. Over 10,000 seventeen-year-olds were asked about their participation in anti-social or risky behaviours. For SES (socio-economic status), the survey’s details around household income, and the educational attainment and occupational status of respondents’ parents were used. A latent measure was extracted from assessments of cognitive ability. Personality measures—the ‘Big Five’—were included, as was a composite measure of hyperactivity. SES and cognitive ability were very weakly associated with anti-social and risky behaviour, while personality measures, and hyperactivity were more strongly linked. Hyperactivity, Agreeableness and Extraversion were the most important measures linked to a measure of anti-social and risky activities (ASRA) and its subscales.

## 1. Introduction

Youthful offending is a predictor of later offending (Andrews and Bonta, 2010) [1], so identifying the causes, or at least correlates, of adolescent criminal and anti-social behaviour is a common research objective. The evidence from the literature around four potential correlates (sometimes referred to negatively as ‘risk’ factors, or positively as ‘protective’ factors) of an individual’s anti-social and offending behaviour is outlined.

**Socio-Economic Status (SES)**—“No single variable has been more important in criminological theorizing than social class… the social origins of crime were in being lower-class, deprived, poor, and frustrated in trying to acquire what the upper classes have” (Andrews and Bonta, 2010: 184). It is almost axiomatic among criminologists that poverty causes crime, despite the paucity of evidence—“the linkage of poverty and crime is inexorable, despite the inability of researchers to establish it at the individual level” (Short, 1991: 501) [2], “social inequality is the main cause of crime” (DeKeseredy and Schwartz, 1996: 463) [3]. However, when Tittle, Villimez and Smith (1978) [4] carried out the first meta-analysis of studies (*n* = 35) examining the class-crime link, they found only a very modest average effect size of −0.09. Reviews and further tests since then (Loeber and Stouthamer-Loeber, 1987; Tittle and Meier, 1990, 1991; Simourd and Andrews, 1994; Gendreau, Little and Goggin, 1996; Dunaway, Cullen, Burton, and Evans, 2000, and Ring and Svenson, 2007) [5,6,7,8,9,10,11] continue to point to a very weak link between and class and crime. Nonetheless, the view is still widely held that low SES and crime must be closely linked for individuals (Webster and Kingston, 2014) [12]. Partly this is due to the impression that there are high-crime neighbourhoods and that these tend to be poor. However, this reflects both the operation of the ecological fallacy (the belief that aggregated crime rates in a large community informs individual behaviour), and the possibility of false causal attribution (the assumption that poverty creates crime, rather than the possibility that criminal behaviour can lead to poverty).

**Cognitive Ability**: Low intelligence has been implicated as a risk factor for offending, violence and anti-social behaviour (Ellis and Walsh, 2003) [13]. Stattin and Klackenberg-Larsson (1993) [14] found that low IQ measured at age 3 predicted crime records up to age 30 controlling for social class: frequent offenders had an average IQ of 88 against an IQ of 101 for non-offenders. Hirschi and Hindelang (1977) [15] found that delinquents scored on average eight IQ points below non-delinquents, and claimed it was as strong a factor as social class in predicting offending. Schweinhart, Barnes and Weikart (1993), Lipsitt, Buka, and Lipsitt (1990) and Denno (1990) [16,17,18] reported low intelligence measured in early childhood predictive of juvenile delinquency and arrests up to respondents’ mid-twenties. Farrington’s longitudinal Cambridge Study of over four hundred boys in a deprived part of London found low intelligence predicted adult convictions, aggression and bullying, spousal assault, and chronic offending (Farrington; Farrington and West, 1993) [19,20,21,22]. These relationships were independent of the effects of family income. The link between low intelligence in crime has been attributed to poor ability to manipulate abstract thoughts and thus to reasoning about the consequences of behaviours; an inability to empathise with others such as victims of crime; weak planning and concentration; and greater experience of school failure, and thus truancy, linked to possible patterns of crime. The inverse of low IQ as a risk factor is the claim that high intelligence can act as a protective factor against offending. Stattin, Romelsjo and Stenbacka (1997) [23] supported this claim in their study of Swedish army conscripts. In a meta-analysis of fifteen studies examining the role of intelligence in offending, Ttofi et al. (2016) [24] found that for individuals deemed at “high-risk” (e.g., family adversity) of being involved in crime, above-average intelligence functioned as a protective factor.

The range of associations between different expressions of anti-social behaviour and cognitive ability is wide (Loeber et al., 2012) [25]. Schwartz and Beaver (2018) [26] in a longitudinal analysis showed that lower cognitive ability was associated with greater likelihood for arrest even controlling for impulsivity. In their analysis of prison inmates, Silver and Nedelec (2018a) [27] found a relationship among prison inmates’ level of cognitive ability, and their misconduct even while in prison. Silver and Nedelec (2018b) [28] also noted more nuanced relationships—cognitive ability acted as a moderating factor for the relationship between anti-social behaviour and neighbourhood disadvantage.

**Personality:** Eysenck (1996) [29] linked personality to criminality, with personality reflected in three main traits. His model is now generally understood as identifying impulsiveness as key to offending—which is described in more detail below. Since 1990 the most widely accepted personality model has been the “Big-Five” or five-factor model (FFM-McCrae and Costa, 1997; 2003) [30,31]. These FFM traits—openness, conscientiousness, extraversion, agreeableness and neuroticism—are construed as biologically based tendencies largely unaffected by environmental influences. Several studies have linked the traits of agreeableness and conscientiousness (negatively) with psychopathy (Hart and Hare, 1994) [32] and self-reported delinquency (Heaven, 1996; John, Caspi, Robins, Moffitt, and Stouthamer-Loeber, 1994) [33,34]. More recently, the link between agreeableness and conscientiousness (again negatively) to antisocial, aggressive and offending behaviour was supported by the work of Jones, Miller, and Lynam, 2011 and Ruiz, Pincus and Schinka, 2008 [35,36]. Walters (2018) [37], focusing on a sample of high-risk delinquent youth—in their late teens—found that agreeableness but not conscientiousness was prospectively predictive of desistance from crime over a five year period. Other research has promoted the idea that personality traits like agreeableness and conscientiousness are important, but require the presence of an intervening variable, such as exposure to a deviant peer group, to have a significant effect (Wilcox, Sullivan, Jones, and van Gelder, 2014; Walters 2018) [38,39].

**Impulsiveness/Hyperactivity:** Impulsiveness appears as a consistently important personality dimension that predicts involvement in crime. However, it is presented in a wide range of constructs along with impulsiveness itself, such as hyperactivity, restlessness, clumsiness, poor planning, acting without planning, and low-self-control. The link between the underlying concept and offending has been explained as due to deficits in the executive functions of the brain (Moffitt, 1993) [40], or alternatively low physiological arousal levels causally associated with sensation-seeking, violence, or impulsive actions. Partial support for this comes from the data showing that offenders tend to have low autonomic levels of arousal such as heart rate or blood pressure (Raine, 1993) [41]. Brennan, Mednick and Mednick (1993) [42] found that ‘hyperactivity’, operationalised as restlessness and poor concentration, and assessed in boys in their early teens, could predict arrests for violence in the cohort up to their early 20s. Bor, McGee and Fagan (2004) [43] linked risk of delinquency at age 14 with assessments in boys of their restlessness and poor attention at age five. A number of Swedish studies linked violence in adult males with assessed restlessness and poor concentration in their teens (Klinteberg, Andersson, Magnusson, and Stattin, 1993; Eklund and Klinteberg, 2003). [44,45] Miller, Flory, Lyman and Leukefeld (2003) [46] found that different elements of impulsiveness such as poor self-control, poor planning, poor perseverance and high sensation-seeking were linked to aggression, drug use and antisocial acts. Sibley et al. (2011) [47] reported that in a longitudinal study of boys, all those with a diagnosis of ADHD were significantly more likely to engage in offending in later years. However, a later study in which nearly 200 boys were assessed at around age 7 on conduct problems, hyperactivity and emotional problems, and followed up to their mid-twenties found that while conduct problems predicted offending in general, and emotional problems predicted arrests for violent offending, hyperactivity measures did not significantly explain any of the variance in offending generally or in specific domains (Young, Taylor, and Gudjonsson, 2016) [48]. Some recent work has used meta-analytic research to examine different elements of impulsivity that may provide a pathway to crime and/or psychopathology, see Berg, Latzman, Bliwise, and Lilienfeld (2015) [49].

The purpose of this paper is to report on an analysis of the self-reported anti-social and risky behaviours of a large representative sample of teenagers, and examine the linkages to their socio-economic status, cognitive ability, personality variables, and hyperactivity (This study was not pre-registered).

## 2. Methodology

### 2.1. The Dataset and Participants

The Millennium Cohort Study (MCS) is a longitudinal study of the cognitive and socio-emotional development of UK children (University of London, Institute of Education, Centre for Longitudinal Studies, 2021) [50]. The MCS is funded by the UK Economic and Social Research Council (ESRC), and several UK government departments. The sample was obtained through a stratified cluster design, and is nationally representative of UK children, with survey weights provided to adjust for non-response, and to enhance representativeness.

The MCS sought to adequately represent disadvantaged children, ethnic minority children, as well as those living in all four countries of the UK. The population was therefore stratified into four countries (England, Wales, Scotland and Northern Ireland), and then into further strata of ‘ethnic minority’, ‘disadvantaged’ and ‘advantaged’ based on data drawn from the electoral ward.

The data are gathered in face-to-face interviews with parents and/or the children. The first wave of the survey was in 2000 when the children were nine months old, and there have been six further waves, at ages 3, 5, 7, 11, 14, and 17. The first wave gathered data from 18,819 children. At age 17, the sample size had declined to 10,757 children, due mainly to attrition. The sample comprised 47.4% males and 52.6% females.

### 2.2. Measures

Univariate statistics (*n*, mean, standard deviation,) for all variables used, are presented at the bottom of Table 1.

#### 2.2.1. Anti-Social and Risky Activities (ASRA)

Wave seven of the MCS asked respondents (aged 17) about a number of behaviours -risky or anti-social—that they had undertaken. Due to the potential sensitivity of the items, this section of the survey was completed using computer-assisted self-interviewing (CASI)—i.e., by the respondent privately on a computer tablet provided by the survey administrator, see Fitzsimons et al. (2020) [51]. They were assured that their answers were confidential, and neither the survey administrator nor the respondent’s parents would have access to the responses. Twenty-three variables were coded dichotomously to create a measure of ASRAs (anti-social and risky activities)—a score of one point for each activity the respondent agreed they had undertaken. For a behaviour to be included, at least 1% of the sample had to agree they had undertaken the activity. Given the chronic nature of some anti-social patterns versus the occasional nature of others, different time frames were employed in the survey for the ASRA elements—last four weeks, last year, lifetime. The selected 23 activities (and % agreeing they had committed the behaviour) were as follows:

##### Substances

1.I sometimes or usually smoke. (10.9%)2.I have had alcoholic drinks 20 or more times in the last twelve months. (19.9%)3.I have used cannabis more than once. (18.1%)4.I have tried ecstasy. (6.2%)5.I have tried cocaine. (5.0%)

##### Anti-Social Activity

6.In the last twelve months, I have shop-lifted. (5.7%)7.In the last twelve months, I have spray-painted or written on property. (2.9%)8.In the last twelve months, I have deliberately vandalised property. (3.1%)9.In the last twelve months, I have set fire to something I should not have. (3.0%)10.In the last twelve months, I have used someone else’s credit/debit card illegally. (1.1%)11.In the last twelve months, I have hacked into someone else’s computer account. (2.0%)12.In the last twelve months, I have hit or punched someone. (23.8%)13.In the last twelve months, I have hit someone with a weapon. (1.0%)14.In the last twelve months I have stolen from somebody. (1.6%)15.In the last twelve months, I have harassed someone online. (1.3%)16.I have carried a knife (while out of home). (1.9%)

##### Gambling

17.In the last four weeks, I have gambled on ‘fruit machines’. (3.7%)18.In the last four weeks, I have placed a bet with other people. (5.4%)19.In the last four weeks, I have placed a bet in a betting shop. (2.1%)20.In the last four weeks, I have placed a bet online. (1.7%)

##### Police Interaction

21.I have been stopped and questioned by the police. (11.6%)22.I have been given a formal warning or caution by the police. (4.7%)23.I have been arrested and taken to a police station. (1.1%)

In total, 10,127 respondents answered these questions. The distribution ranged from zero activities to 19 activities and was highly skewed, with 46.9% reporting zero activities, and one individual scoring nineteen, and with an overall mean of 1.4. Of the sample, 12.7% had a score of 3 or higher.

The overall ASRA scale was also broken down into four separate sub-scales reflecting different families of possible deviant behaviours. These were grouped into the domains listed above: substances, anti-social acts, gambling, police interaction.

The distribution of the dependent variable, ASRA, was highly skewed, and modelled as count data, with a low mean, no negative values, a mode of 0, highly asymmetric and the data stacked towards 0. Therefore, to examine the role of multiple potential predictors of ASRA, a Poisson regression was used. Data around offending such as population criminal activity, or victimisation are typically well-modelled by the Poisson distribution, see Prieto-Curiel, Collignon-Delmar and Bishop (2018), and Maltz (1996) [52,53]. The relative importance of the predictors was indicated by the size of the Wald chi-square.

#### 2.2.2. Cognitive Measures—Child’s Cognitive Ability

The MCS included cognitive ability tests at waves 3, 4, 5 and 6 (Rosenberg, Atkinson, Abdullah, Agalioti-Sgompou, 2020) [54]. A single unrotated principal component score was extracted for each participant based on the following: 

Wave 3, age 5—iPattern Construction T-score, Naming Vocabulary T-score, Picture Completion T-score.

Wave 4, age 7—Word Reading Standard Score, Pattern Construction T-score, Maths 7 Standardised Age Score.

Wave 5, age 11—Verbal Sims Standard Score.

Wave 6, age 15—Word Activity (Recognition) Score out of 20.

Data (non-imputed) were available for 7428 respondents for all eight cognitive measures.

#### 2.2.3. Socio-Economic Status (SES) Measures

##### Household Income

A derived variable from each survey wave measuring weekly household income based on net earnings, benefits, and pensions was provided by the MCS. This was modified (by the MCS) using the OECD equivalence scale of a value of 1 for the first adult, 0.7 for each adult after that and 0.5 for each child in the household. The estimate for household income for wave 6 was used (child aged 15) so that it was temporally prior to the collection of ASRA data in wave 7.

##### Highest Parental Education

The MCS provided a measure of highest educational attainment of either parent based on a scale from 1–5 (lowest to highest). The data from wave 6 were used. Parents at level 1 had primary level education plus a small amount of secondary education, whereas parents at level 5 had at least one university degree.

##### Highest Parental Occupational Status

The MCS recorded the job status of both parents using the SOC-2000 coding scheme. These were recoded into categories ISCO-88, and the ISCO-88 categories were assigned a ISEI occupational status score. The ISEI is a worldwide scale that transforms narrow occupation roles into a numerical scale based on the transmission of educational qualifications into earnings through occupation. It maximizes the importance of occupation on income, net of education (Ganzeboom and Treiman, 1996) [55]. The highest occupation status of either parent was assigned. *n* = 4528, and the minimum score was 16, and maximum was 88. Data from wave 6 were used.

#### 2.2.4. Hyperactivity

In wave 7, hyperactivity was assessed with a sub-scale of the Strengths and Difficulties Questionnaire (SDQ—see Goodman, 1997) [56]. This sub-scale comprised of five items respondents answered about themselves on a 3-point scale: “I am restless, I cannot stay still for long”, “I am constantly fidgeting or squirming”, “I am easily distracted, I find it difficult to concentrate”, “I think before I do things” (Reversed), “I finish the work I am doing. My attention is good.” (Reversed).

The reliability (Cronbach’s Alpha) for the five items was 0.69, *n* = 9799.

#### 2.2.5. Personality Measures

In wave 7 of the MCS, the “Big Five” personality traits-Openness (*n* = 9885), Conscientiousness (*n* = 9876), Extraversion (*n* = 9865), Agreeableness (*n* = 9889), Neuroticism (*n* = 9881)—of the cohort, aged 17, were assessed with five three-item scales. The reliability scores for four of the five traits were low (respectively, 0.68, 0.60, 0.67, 0.63, 0.79)—between 0.6 and 0.70—ideally these should all be above 0.7 (DeVellis, 2003) [57], but Pallant (2010; 97) [58] notes that “for scales with fewer than ten items, it is common to find quite low Cronbach values., e.g., 0.5”. In Appendix A, the items used to measure each of the personality traits are presented.

### 2.3. Imputation of Data

As most measures had missing values, imputation of values was employed, using the ‘Fully Conditional Specification MCMC’ command in SPSS 26. No imputation was used on the ASRA measure (as this was the dependent variable—assessing anti-social and risky behaviour). The variables which had most missing values prior to imputation were the child’s cognitive score (*n* = 7428) and parental occupational status (*n* = 4528). Weights were applied. Following imputation and weighting, *n* = 8338. In the regression analysis, all independent measures—except gender—have been standardised.

## 3. Results

The correlation matrix for the dependent measure, anti-social and risky activity (ASRA) and ten potentially associated measures is presented in Table 1. The strongest correlation with ASRA was the measure of hyperactivity (r = 0.25). Other variables with moderate associations with ASRA were the personality measures of conscientiousness (negatively at r = −0.12), extraversion (r = 0.12) and agreeableness (negatively at r = −0.18). Due to the large sample size, there are statistically significant associations between ASRA measures of SES such as highest parental education value, and household income. However, these associations are in reality very small. Similarly, the associations between ASRA and the cognitive ability measure, as well as the personality trait of ‘openness’, while statistically significant, are very low. There was no significant association between the personality trait of neuroticism or the SES measure of parental occupation, and ASRA.

The overall model (dependent measure is ASRA, and entering eleven independent variables) had a likelihood ratio-chi square (pooled among five sets of imputations) = 2732, which was significant < 0.001. In Table 2, the Wald chi-square values for each of the independent measures in the model is shown, along with its significance, the B value, and the ExpB (the exponentiated value of B is the predicted change in an ASRA score of 1.000 for every change of one standardised unit of the independent measure).

The sub-scales of the ASRA measures (substances, anti-social acts, gambling, police interaction) were also assessed separately. The results of the Poisson regressions are presented in Table 3 below.

As can be seen in Table 3, the domains of substances, and of anti-social activity were better explained overall than were those of gambling, and police interaction. Hyperactivity was a key and substantial predictor for three of the domains, though not for gambling. Both extraversion and agreeableness (negatively) were also important predictors for three of the domains, but not for police interaction. While cognitive ability does appear to have modest links with two of the domain outcomes, this is in a counter-intuitive way—with higher ability linked to more offending. None of the variables around SES had significant associations with the domains, except in the case of police interaction, where lower household income was associated with more interaction.

## 4. Discussion

### 4.1. Summary

The key area of enquiry of this research was on the correlates of anti-social and risky behaviour among a large representative group of teenagers. These correlates are often presented causally as ‘risk’ or alternatively as ‘protective’ factors. In wave seven of the MCS, over 10,000 seventeen-year-olds were asked about their participation in a range of anti-social or risky behaviours (ASRA) in the areas of drinking alcohol, smoking, trying a number of illegal drugs; general anti-social behaviour such as stealing, vandalising, physical aggression; gambling, as well as coming into contact with the police. Information was gathered on the socio-economic status (SES) of the respondent such as their household income, as well as the educational attainment and occupational status of their parents. A composite measure of the cohort’s cognitive ability was derived from calculating the first principal component of eight separate measures gathered from four previous waves of the longitudinal study. Personality measures—specifically the Big Five traits, as well as the hyperactivity subscale of the SDQ—were also included. The analysis showed that SES and cognitive ability were very weakly associated with anti-social and risky behaviour, while personality measures were more strongly linked. In a Poisson regression, only the personality measures were significant predictors of ASRA, and SES measures and cognitive ability explained very little. Hyperactivity, Agreeableness and Extraversion were the most important personality measures linked to ASRA. When ASRA was broken down into four sub-scales or domains, hyperactivity remained key for three of the domains, as did agreeableness and extraversion. An SES variable, household income (negatively) predicted interaction with the police.

### 4.2. Concordance with the Literature

The link between personality measures and anti-social, risky or criminal behaviour is well established. As noted above, impulsivity in childhood and/or adolescence—defined in a number of ways, but similar to the construct of hyperactivity—has consistently been found to be predictive in prospective studies, of later arrests and criminal involvement. Traits in the Big Five were also associated with criminal and anti-social behaviour, particularly that of Agreeableness (negatively). However, the relative weakness of Conscientiousness and relative strength of Extraversion represented differences from the main literature. It is possible that being extraverted might lead one to spend more time in the company of others, therefore increasing potential opportunity to some of the ASRA items like aggressing against others. It might also be conducive to more alcohol, tobacco or drug consumption, and to gambling as a social or group activity. The correlation between Conscientiousness and Hyperactivity was quite strong at r = −0.42, and the items measuring hyperactivity seem to tap into some of the construct of what it means to be conscientious, and this may explain why Conscientiousness was a very weak predictor in the multiple regression, once Hyperactivity was included.

The weakness of SES is not surprising. As has been noted in other domains, while many social scientists assume SES provides the decisive backdrop to people’s lives, its relationship to anti-social behaviour and crime—when actual data are gathered allowing a sober assessment—usually proves to be very minor.

More surprising was the very weak link between cognitive ability and the ASRA measure. A far stronger link has been reported and replicated in many studies. However, many of the studies reporting a link have been of relatively small samples deemed to be at high risk (based on family adversity or early involvement in crime) compared to matched controls—for example, Bender et al. (1996), Farrington, Ttofi, and Piquero, (2016), Jaffee et al. (2007), Loeber, Pardin, Stouthamer-Loeber, and Raine (2007), or Werner and Smith, (1982) [59,60,61,62,63]. The other line of research studies reporting a link between cognitive ability have been of much larger representative families, but where data on anti-social and offending behaviour has continued into adulthood, for those in their late 20s or even thirties—for example, Klika, Herrenkohl, and Lee (2012) [64]. Thus, it is feasible that cognitive ability is important as a ‘protective’ factor among somewhat atypical at-risk adolescents, and that it is also of importance to distinguish in the general population between those who continue to transgress into adulthood and those who desist as they reach adulthood.

### 4.3. Limitations

The key measure was dependent upon the accuracy and honesty of the respondents’ self-reported activities. The counting of activities and use of the same score for actions of potentially very different levels of seriousness (e.g., carrying a knife, or being arrested, versus gambling with friends or alcohol consumption) inevitably raises measurement validity concerns. The measure was highly skewed with almost half the sample reporting zero ASRAs, and a long tail up to an individual reporting nineteen of them. Other standard problems around longitudinal studies also hold—the attrition of the cohort over time, usually non-randomly. Nonetheless, this was a large-scale representative national sample of teenagers being assessed over time, and providing information around the sensitive issue of their involvement in either anti-social, risky or illegal behaviours.

### 4.4. Implications

The findings suggest a number of rich areas for further research. The role of cognitive ability needs to be more fully investigated, potentially as important for specific atypical groups in adolescence, and then as a possible more general protective factor: against turning transgressions in adolescence into a chronic pattern of recidivism in adulthood. The operationalisation of ‘at-risk’ may also have to be reviewed. Personality factors have again proven more decisive than SES in distinguishing between young people who are more or less problematic in terms of their anti-social behaviour. Rather than categorising people as at-risk due to adversity in the sense of poverty, it might be more meaningful and parsimonious to look for consistent markers of impulsivity and poor levels of agreeableness among young people as significant indicators of future chronic offending behaviour.

## Figures and Tables

**Table 1 behavsci-13-00046-t001:** Correlation matrix of continuous measures, imputed data. *n* = 10,080. Pooled r scores based on five imputations. *p* < 0.05 * *p* < 0.01 **.

Measure	1	2	3	4	5	6	7	8	9	10	11
1. **Anti-social or risky activities measure**	1										
2. **Child cognitive ability**	0.03 **	1									
3. **Highest parent educate**	0.03 **	0.26 **	1								
4. **Household income**	0.03 *	0.40 **	0.46 **	1							
5. **Highest parent occup.**	0.02	0.23 **	0.42 **	0.36 **	1						
6. **SDQ-Hyperactivity**	0.25 **	−0.01	−0.03 *	−0.06 **	−0.03 *	1					
7. **Pers.—Openness**	0.03 **	0.15 **	0.09 **	0.10 **	0.08 **	−0.05 **	1				
8. **Pers.—Conscientious**	−0.12 **	0.10 **	0.05 **	0.09 **	0.04 **	−0.42 **	0.30 **	1			
9. **Pers.—Extraversion**	0.12 **	0.02	0.03 *	0.07 **	0.08 **	−0.05 **	0.22 **	0.26 **	1		
10. **Pers.—Agreeableness**	−0.18 **	0.04 **	0.05 **	0.06 **	0.07 **	−0.28 **	0.34 **	0.43 **	0.23 **	1	
11. **Pers.—Neuroticism**	−0.01	0.05 **	0.00	0.04 **	−0.01	0.23 **	0.09 **	−0.11 **	−0.18 **	0.10 **	1
*n (non-imputed)*	10,127	7428	9343	9303	4528	9799	9885	9876	9865	9889	9881
*Mean (non-imputed)*	1.40	−0.09	3.28	423.02	47.65	3.94	14.18	14.10	13.50	16.57	11.79
*Standard Deviation* *(non-imputed)*	2.07	1.03	1.09	179.3	19.31	2.32	4.08	3.62	4.13	3.43	4.92

**Table 2 behavsci-13-00046-t002:** Poisson regression with ASRA as the dependent measure, and eleven independent measures. *n* = 8338. Estimates based on the pooled means of five imputations. All independent variables standardised except for gender. *Non-significant predictors—above < 0.001—are italicised*.

Measure	B	Wald Chi-Square	Exp (B)	Sig. Level
Hyperactivity	0.292	801.6	1.339	<0.001
Personality-Extraversion	0.239	605.6	1.271	<0.001
Personality-Agreeableness	−0.221	513.4	0.802	<0.001
Personality-Openness	0.108	117.6	1.091	<0.001
Cognitive Ability	0.062	39.7	1.064	<0.001
Personality-Neuroticism	−0.032	12.7	0.968	<0.001
*Parental education attainment*	*0.029*	*10.5*	*1.030*	*=0.011*
*Personality-Conscientiousness*	−*0.026*	*5.9*	*0.975*	*=0.017*
*Equivalised household income*	−*0.020*	*4.1*	*0.981*	*=0.048*
*Parental occupational status*	*0.003*	*2.0*	*1.003*	*=0.130*
*Gender*	−*0.005*	*0.2*	*0.995*	*=0.593*

**Table 3 behavsci-13-00046-t003:** Poisson regression with four sub-scales of the ASRA—substances, anti-social behaviour, gambling, police interaction—and eleven independent measures. *n* = 8338. *Non-significant predictors (where sig. is not < 0.001) are italicised*.

**Substances Sub-Scale, (5 Items)** Likelihood Ratio Chi-Square = 1428.7				
**Measure**	**B**	**Wald Chi-Square**	**Exp (B)**	**Sig. Level**
Personality-Extraversion	0.320	472.7	1.377	<0.001
Hyperactivity	0.275	322.2	1.316	<0.001
Personality-Agreeableness	−0.198	185.9	0.820	<0.001
Cognitive Ability	0.086	32.8	1.091	<0.001
Personality-Conscientiousness	−0.081	26.6	0.922	<0.001
Personality-Openness	0.068	21.2	1.070	<0.001
*Parental education attainment*	*0.051*	*11.1*	*1.052*	*=0.001*
*Gender*	*0.055*	*4.3*	*1.057*	*=0.049*
*Equivalised household income*	*0.005*	*1.5*	*1.006*	*=0.183*
*Parental occupational status*	*0.004*	*2.5*	*1.004*	*=0.136*
*Personality-Neuroticism*	*−0.001*	*0.1*	*0.999*	*=0.921*
**Anti-Social Sub-Scale, (11 items)** Likelihood Ratio Chi-Square = 1466.4				
**Measure**	**B**	**Wald Chi-Square**	**Exp (B)**	**Sig. Level**
Hyperactivity	0.384	481.8	1.467	<0.001
Personality-Agreeableness	−0.305	359.7	0.737	<0.001
Personality-Openness	0.205	143.4	1.228	<0.001
Personality-Extraversion	0.157	94.5	1.170	<0.001
Cognitive Ability	0.083	25.7	1.087	<0.001
Gender	−0.0116	13.9	0.890	<0.001
*Equivalised household income*	*−0.054*	*10.1*	*0.948*	*=0.001*
*Parental education attainment*	*0.054*	*9.9*	*1.055*	*=0.001*
*Personality-Neuroticism*	*−0.043*	*7.9*	*0.958*	*=0.004*
*Parental occupational status*	*−0.019*	*3.9*	*0.982*	*=0.067*
*Personality-Conscientiousness*	*0.019*	*1.1*	*1.019*	*=0.290*
**Gambling sub-scale, (4 items)** Likelihood ratio chi-square = 160.9				
**Measure**	**B**	**Wald Chi-Square**	**Exp (B)**	**Sig. Level**
Personality-Extraversion	0.235	43.2	1.266	<0.001
Personality-Neuroticism	−0.170	25.7	0.843	<0.001
Personality-Agreeableness	−0.163	20.5	0.850	<0.001
Parental education attainment	−0.139	15.3	0.871	<0.001
*Personality-Conscientiousness*	*0.129*	*11.2*	*1.138*	*=0.001*
*Equivalised household income*	*0.095*	*9.9*	*1.103*	*=0.002*
*Cognitive Ability*	*−0.093*	*8.4*	*0.912*	*=0.003*
*Hyperactivity*	*0.081*	*4.9*	*1.084*	*=0.030*
*Parental occupational status*	*0.040*	*4.9*	*1.043*	*=0.047*
*Personality-Openness*	*0.047*	*1.9*	*1.048*	*=0.173*
*Gender*	*0.044*	*0.5*	*1.045*	*=0.452*
**Police interaction sub-scale, (3 items)** Likelihood ratio chi-square = 180.6				
**Measure**	**B**	**Wald Chi-Square**	**Exp (B)**	**Sig. Level**
Hyperactivity	0.263	71.3	1.301	<0.001
Equivalised household income	−0.117	14.5	0.890	<0.001
*Cognitive Ability*	*0.007*	*6.5*	*1.010*	*=0.010*
*Parental education attainment*	*−0.025*	*5.3*	*0.977*	*=0.021*
*Personality-Agreeableness*	*−0.038*	*3.9*	*0.963*	*=0.053*
*Parental occupational status*	*0.039*	*2.5*	*1.040*	*=0.126*
*Personality-Openness*	*0.039*	*2.0*	*1.040*	*=0.135*
*Personality-Neuroticism*	*−0.037*	*1.8*	*0.964*	*=0.193*
*Personality-Extraversion*	*0.182*	*1.5*	*1.199*	*=0.222*
*Gender*	*0.023*	*0.2*	*1.023*	*=0.624*
*Personality-Conscientiousness*	*0.000*	*0.1*	*1.000*	*=0.893*

## Data Availability

Dataset available via Centre for Longitudinal Studies, UK.

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
