# Peer review of "Assessing Patterns of Anti-Social and Risky Behaviour in the Millennium Cohort Study—What Are the Roles of SES (Socio-Economic Status), Cognitive Ability and Personality?"

_behavsci, 2023, doi:10.3390/bs13010046_

Round 1

Reviewer 1 Report

Dea author.
It's been a long time since I read an article as complete and clear as yours. I found in your work an excellent methodology, a deep bibliographical review, relevant results and a discussion according to what the reader expects to find. Thank you very much for your contributions to science.
Congratulations!

Author Response

Thank you for your very kind, positive and supportive comments. 

Reviewer 2 Report

The Introduction of the manuscript lacks a more elaborate construction.

The authors use the argument of the importance of investigating risk factors and, from then on, explain about certain variables/constructs, rescuing some literature, to justify their choice and investigation in the reported study.

Some of the variables chosen to be tested, however, do not have sufficient support in the literature of risk factors; they do not appear in the results of recent (or old) meta-analyses. 

This is the case, for example, of the Cognitive Ability. As well as the Socio-Economic Status (SES) variable. 

Regarding SES, the defence of this as a risk factor is the most problematic. Since the studies of Ivan Nye, in 1958, the role of this variable has been questioned and, as the author of the manuscript himself affirms: "It is almost axiomatic among criminologists that poverty 30 causes crime, despite the paucity of evidence...".

Thus, it becomes clear that the option for the investigation of the variables has more to do with the data to which He has access and with an exploratory perspective. 

It would be very interesting if the author explained the variables that make up the database in the different waves of the survey and why he "chose" the privileged variables.

Testing each of them is interesting, especially in a representative sample. But their "choice" is not justified by the argument that they have figured as a risk factor (with the exception of "personality" and "impulsiveness"). It would be important to build another argument for the importance of testing Cognitive Ability and SES. 

Within this questioning, it would be very interesting to know if SES predicts detentions (if there is this data in the study database and if the researcher can access it). This test would bring an important and additional contribution.

Furthermore, the self-reported delinquency rates found in the studies were very low, as the author himself states. It would be important to provide in the text a comparison of these rates with those of other prevalence studies - also based on self-reported delinquency - and question/problematize further why, despite the care taken with data collection in this regard the rates are small (lower than those found in other studies in the UK).

Author Response

Thank you to Reviewer 2 for the detailed comments on the paper.

The reviewer is sceptical of the rationale for the selection of the risk factors that have been assessed in relation to the dependent variable – anti-social and risky activities (ASRA). The possibility that the independent variables chosen were simply ones of convenience from the Millennium Cohort Study (MCS) is suggested.

However, the MCS is a very large survey and has gathered data on many aspects of young people’s lives over six sweeps. These measures are extremely diverse, and range from physical health, to mental health, time use, educational achievements, family life, attitudes and beliefs, social connectedness, religion, financial situation, biomedical data, cognitive ability, leisure, peer relations. There were many other variables that could have been selected for analysis. The ones that were chosen were picked not for convenience, or because they were the only ones, but for their relevance to the question at hand. See https://cls.ucl.ac.uk/cls-studies/millennium-cohort-study/.

The reviewer seeks justification in particular for including cognitive ability and socio-economic status (SES) as potential risk factors or key correlates for the outcome measure.

The justification for including cognitive ability is based partly on the belief that this paper is part of a special collection of papers looking at the impact of cognitive ability and personality on important life outcomes. Thus, it seems important to include cognitive ability as a possible (negative) risk factor for the outcome. Furthermore, lower cognitive ability has been associated with poorer impulse control, poorer control of inhibition, more planning deficits, and weaker long-term thinking. Therefore if seems reasonable on grounds of both coherence with the special edition of papers, as well as on conceptual and evidential grounds to include, control for and assess the role of cognitive ability in these anti-social and risky activities.

In relation to SES, it is contended (and evidence is provided) that there is a widespread belief, particularly in sociological theorising on the causes of different forms of crime and anti-social activity that SES is a variable of great importance. A number of citations were listed making this point. The reviewer is correct that meta-analyses supporting this view is absent – but the view remains dominant in neo-Marxist theories of crime. The problem is that this view is maintained in the absence of any supporting data, probably for political/ideological reasons. If SES had not been included in the analysis, it is quite likely that any critique of it would have contended that had SES been included, most of the variance in the outcome measure would have been explained by it. Therefore it was felt important to include it, and control for its (very slight) effects.

Reviewer 3 Report

This well-written, tightly-focused paper uses data from the >10,000 17-year-olds included in the Millenium Cohort Study, to test for correlates of anti-social and risky behavior by adolescents. Measures of SES, cognitive ability, and personality, including hyperactivity were tested as correlates of self-reported anti-social and risky behavior, finding the strongest correlation with hyperactivity and moderate correlation with conscientiousness/agreeableness (negatively) and with extraversion. Links with cognitive ability were very weak, contrary to the literature, as were links with SES measures. The author points to the need for more research on the role of cognitive ability in predicting anti-social behavior of adolescents and into adulthood.  The findings further support findings in the literature showing weak causality of SES measures compared to negative correlation with personality measures such as agreeableness and conscientiousness.

The paper is very clear and straightforward, and the unique, large longitudinal data base lends itself well to this multivariate analysis.

Author Response

Thank you for you very clear and positive feedback on this paper. 

Reviewer 4 Report

1. SES, ASRA in the title and abstract should be explained.

2. In the introduction, most of the references are over 20 years old - does this mean there is currently no research on the analyzed topic?

3. What was the representativeness of the sample - what are the characteristics of the population?

4. Please describe the sampling methods in more detail.

5. Please describe Statistical methods in the Methods section.

6. Please explain the abbreviations used in the table in the footnotes.

7. Please justify the choice of the regression model. Were other models designed?

8. Please justify setting the significance level to <0.001.

9. Please summarize the results in the conclusions section.

Author Response

Responses to Reviewer 4

  1. SES, ASRA in the title and abstract should be explained.

The abbreviations have been replaced by the full terms in the abstract and title.

  1. In the introduction, most of the references are over 20 years old - does this mean there is currently no research on the analyzed topic?

Much of the influential research into predictors of anti-social behaviour emerged from a number of analyses of key longitudinal datasets of large representative samples of young people, which were published over 20 years ago. And the Introduction does include a number of references that are more recent such as Ring & Svenson (2007), and Webster & Kingston (2014), Ttofi et al, (2016). However, looking again at the literature on cognitive ability, a number of further relevant studies were found, and the following has been added to the second section of the Introduction:

The range of associations between different expressions of anti-social behaviour and cognitive ability is wide (Loeber et al., 2012). Schwartz & Beaver (2018) in a longitudinal analysis showed that lower cognitive ability was associated with greater likelihood for arrest even controlling for impulsivity. In their analysis of prison inmates, Silver & Nedelec (2018a) found a relationship among prison inmates’ level of cognitive ability, and their misconduct even while in prison. Silver & Nedelec (2018b) also noted more nuanced relationships – cognitive ability acted as a moderating factor for the relationship between anti-social behaviour and neighbourhood disadvantage.

  1. What was the representativeness of the sample - what are the characteristics of the population?

The data are drawn from a longitudinal study, the Millennium Cohort Study (MCS). When started in the year 2000, it consisted of a representative sample of all the children born in the UK in that year. Because of attrition, the sample had declined by wave 7 to 10,757 children (47.4% male, and 52.6% female) aged 17. Weighting, as is the standard approach – was used to ensure continued survey representativeness. These details are now fully outlined in section 2.1.

  1. Please describe the sampling methods in more detail.

The following has been added to section 2.1:

The MCS sought to adequately represent disadvantaged children, ethnic minority children, as well as those living in all four countries of the UK. The population was therefore stratified into four countries (England, Wales, Scotland and Northern Ireland), and then into further strata of ‘ethnic minority’, ‘disadvantaged’ and ‘advantaged’ based on data drawn from the electoral ward.

  1. Please describe Statistical methods in the Methods section.

The following section was moved from the Results section to the Methods section 2.2.1:

The distribution of the dependent variable, ASRA, was highly skewed, and modelled as count data, with a low mean, no negative values, a mode of 0, highly asymmetric and the data stacked towards 0. Therefore to examine the role of multiple potential predictors of ASRA, a Poisson regression was used. Data around offending such as population criminal activity, or victimisation are typically well-modelled by the Poisson distribution, see Prieto-Curiel, Collignon-Delmar & Bishop (2018), and Maltz (1996). The relative importance of the predictors was indicated by the size of the Wald chi-square.

  1. Please explain the abbreviations used in the table in the footnotes.

In my version, there is only one footnote, and it does not contain any table or abbreviation? There is an appendix, but this has a list of Likert scales, and no abbreviations? So, it is not clear where the footnotes being referred to are.

  1. Please justify the choice of the regression model. Were other models designed?

As noted in the methods section, the dependent variable was highly skewed with a mode of 0, no negative values, and highly asymmetric, so the assumptions of more standard models, like Ordinary Least Squares regression would not have been appropriate. With a Poisson distribution, the Wald chi-square was preferred.

  1. Please justify setting the significance level to <0.001.

The number in the sample was very high, at over 10,000. With these large numbers often very insubstantial differences appear as statistically significant. The use of 0.001 as a cut-off norm was deemed therefore to be a small but useful attempt to indicate some difference or effect of substance, rather than just statistical significance.

  1. Please summarize the results in the conclusions section.

Currently the paper does not have a conclusions section, but uses the standard format  followed by the vast majority of scientific journals – IMRAD (Introduction, Methods, Results and Discussion), see Ecarnot et al (2015). But there is a summary of the results available in the first section of the Discussion: section 4.1, Summary.

Round 2

Reviewer 4 Report

Tables must be self-explanatory - all abbreviations (SDQ, sig.) need to be defined either in the table itself or in the notes under the table. And this needs to be repeated separately for each table in a paper.